



*Original Research article*

# Development of soil biological quality index for soils of semi-arid tropics

**Selvaraj Aravindh, Chinnappan Chinnadurai, and Danajeyan Balachandar***

Department of Agricultural Microbiology, Tamil Nadu Agricultural University, Coimbatore 641003, India

Correspondence to: D. Balachandar (dbalu@tnau.ac.in)

Abstract

The Agricultural intensification, an inevitable process to feed the ever-increasing population, affects the soil quality due to management-induced changes. To measure the soil quality in terms of the soil functioning, several
attempts were made to develop the soil quality index (SQI) based on a set of soil attributes. However, there is no universal consensus protocol available for SQI and the role of soil biological indicators in SQI is meagre. Therefore, the objective of the present work is to develop a unitless soil biological quality index (SBQI) scaled between 0 and 10, which would be a major component of SQI in future. The long-term organic manure amended (OM), integrated nutrient management enforced (INM), synthetic fertilizer applied (IC) and unfertilized control
(Control) soils from three different predominant soil types with three different cropping patterns of the location (Tamil Nadu state, India) were chosen for this. The soil organic carbon, microbial biomass carbon, labile carbon, protein index, dehydrogenase activity and substrate-induced respiration were used to estimate the SBQI. Five different SBQI methods viz., simple additive (SBQI-1 and SBQI-2), scoring function (SBQI-3), principal component analysis-based statistical modeling (SBQI-4) and quadrant-plot based method (SBQI-5) were developed to
estimate the biological quality as unitless scale. All the five methods have same resolution to discriminate the soils and INM ≈ OM > IC > Control is the relative trend being followed in all the soil types based on the SBQIs. All the five methods were further validated for their efficiency in 25 farmers' soils of the location and proved that these methods can be effectively used to scale the biological health of the soil. Among the five SBQIs, we recommend SBQI-5, which relates the variables to each other to scale the biological health of the soil.

**Keywords:** Soil health; Soil quality index; Biological indicators; Sustainable soil management

## 1. Introduction

Soil quality, according to Doran and Parkin (1994), is the capacity of a soil to function, within the ecosystem and land use boundaries, to sustain productivity, maintain environmental quality and promote plant and animal health. Soil quality uses several physical, chemical and biological attributes of soil either individually or in
combinations to determine if the soil function under different management and agricultural practices is improving, stable or degrading (Andrews et al., 2002;Andrews et al., 2004;Karlen et al., 2001). As the soil functions of interest and the environmental factors differ among the soil systems, there is no universal methodology available to measure the soil quality using a common set of indicators (Bouma, 2002). Measures of selected soil attributes are used to assess the soil quality which are referred as 'soil quality indicators'. Their measure in the soil as influenced



by nutrient management, tillage, cropping system, and all ecosystem disturbance activities were used to assess the soil quality and its sustainability (Andrews et al., 2004;Karlen et al., 2006;Masto et al., 2008). Alternatively, soil properties such as soil organic carbon and their fractions, soil aggregates and their stability  and several microbial attributes, which are sensitive to management practices were also used to monitor the soil quality (Bastida et al., 2006;Carter, 2002;Li et al., 2013;Masto et al., 2008). Apart from these, several biochemical properties including respiration, nitrification and enzymes' activity were also reported as the good sensitive indicator for soil quality (Bastida et al., 2006;Gil-Sotres et al., 2005;Lopes and Reynolds, 2010).  However, the choice of soil indicators and their contribution to soil quality vary according to several factors including climate, intended land use patterns and so on (Karlen et al., 2006). Soil quality was used as a tool to evaluate the effects of soil management practices and tillage systems (Armenise et al., 2013;Hussain et al., 1999;Shukla et al., 2006), land use type (Masto et al., 2008;Rahmanipour et al., 2014), cover crop (Bastida et al., 2006;Fu et al., 2004;Navas et al., 2011) and native ecosystems and grassland degradation (Alves de Castro Lopes et al., 2013;Li et al., 2013) on soil function.

The term 'soil quality index' (SQI) is defined as 'the minimum set of parameters that, when interrelated, provides numerical data on the capacity of soil to carry out one or more functions' (Acton and Padbury, 1993). The soil quality index is the functions of soil quality indicators, which is defined as 'measurable property that influences the capacity of a soil to carry out a given function' (Acton and Padbury, 1993). The soil quality index assessment studies indicated that SQI is complex due to diversity of soil quality indicators (representing physical, chemical and biological attributes of the soil) and unease to integrate them all to establish in to a single measurable scale (Garcia et al., 1994;Halvorson et al., 1996;Papendick and Parr, 1992). Several attempts were made to find a way to aggregate the information obtained for each soil quality indicator into a SQI. The simple addition of soil quality indicators (Velásquez et al., 2007) or scoring function of soil quality indicators (Moebius-Clune et al., 2016) are the two common approaches used to scale the soil quality index between 0 and 1 or 0 and 10. The selection of soil quality indicators should be deliberating to the soil functions of interest (Nortcliff, 2002); threshold values of such identified indicators should be based the local conditions and indicator selection should be based on experts' opinion or statistical procedures or combination of both to obtain a minimum data set. However, the soil quality index should link the scientific knowledge and agricultural and land management practices in order to assess sustainability (Romig et al., 1995). Most of the SQI give more importance to the physical (soil aggregation, water retention) and chemical indicators (carbon dynamics and nutrient carrying capacity) with less importance to biological attributes (microbial biomass carbon, arthropods) (Biswas et al., 2017;Calero et al., 2018;de Paul Obade and Lal, 2016;Lima et al., 2013;Menta et al., 2018;Paz-Kagan et al., 2014;Pulido et al., 2017;Qi et al., 2009;Raiesi, 2017;Schmidt et al., 2018;Vasu et al., 2016). In order to emphasize the biological and biochemical attributes to soil quality, the biological quality of soil (BSQ) was first proposed by Parisi (2001) which used to measure the bioindicators of soil, especially the arthropods of soil. This approach was successfully validated with other physical and chemical indicators in several works (Aspetti et al., 2010;Begum et al., 2011;Blasi et al., 2013;Garcia et al., 1994;Madej et al., 2011;Mazzoncini et al., 2010;Menta et al., 2018;Menta et al., 2014;Raglione et al., 2011;Rüdisser et al., 2015;Visioli et al., 2013). Pascazio et al. (2018) used microbial biomass, β glucosidase, mineralizable nitrogen and urease to represent the biological indicators to measure the SQI. Similarly, Vincent et al. (2018) used bacterial and fungal density and richness with mycorrhizal colonization as bioindicators for SQI. From these works, it is evident that there is no consensus to represent the biological component of the SQI.

With this background, we have used six important biological attributes of the soil, selected based on our previous works (Balachandar et al., 2016;Balachandar et al., 2014;Chinnadurai et al., 2013;Chinnadurai et al., 2014;Tamilselvi et al., 2015;Preethi et al., 2012) such as soil organic carbon, microbial biomass carbon, labile carbon, protein index, dehydrogenase activity and substrate-induced respiration as soil biological quality indicators. The





actual measures of these were scaled to untiless SQI (between 0 to 10) using five different methods. We have used the term 'soil biological quality index' (SBQI) instead of 'soil quality index', as we focused to use most of the biological, specifically, microbial attributes of the soil to measure the quality omitting physical and chemical indicators. The primary aim of this work is to identify minimum dataset to represent the total biological activities of the soil and its contribution to the soil quality. Hence, SBQI may be a component of SQI in future, after integrating the physico-chemical indicators to it.

## 2. Materials and Methods

### 2.1. Experimental sites and soil sampling

Long-term permanent manurial trials being maintained by Tamil Nadu Agricultural University, India at three different locations of Tamil Nadu state, India viz., Department of Soil Science and Agricultural Chemistry, Coimbatore, Agricultural College and Research Institute, Madurai and Agricultural Research Station, Kovilpatti (designated as Coimbatore, Madurai and Kovilpatti, respectively) were selected for soil quality analysis. The details of long-term permanent manurial trails are described in Table 1. Four long-term nutrient management treatments viz., unfertilized control soil (control); inorganic fertilizers in prescribed recommended dosage applied soil (IC); organic amendments (farm yard manure) in the dose of N-equitant basis applied soil (OM) and integrated nutrient management (both organic and inorganic) adopted soil (INM) were chosen for this study. Top soils (0-25 cm) were collected from these plots when the crop is not raised (January, 2018). Each sample was the composite of 10 random soil cores from each plot after thoroughly mixed and nine such replicates were maintained per soil. The soil samples were placed in plastic bags, transported to the laboratory, homogenized and stored at 4°C. The gravimetric moisture content of the soil was measured immediately.

### 2.2. Soil biological properties

Soil organic carbon (SOC) was analyzed by wet chromic acid digestion method (Walkley and Black, 1934) and expressed as mg per g of soil. The microbial biomass carbon (MBC) was measured by fumigation-incubation technique (Jenkinson and Powlson, 1976) and expressed as μg per g of soil. Soil labile carbon (SLC) was measured by the permanganate method (Blair and Crocker, 2000) and expressed as μg per g of soil. Soil protein was extracted from soil using a protocol as described by Hurisso et al. (2018) and expressed as μg per g of soil. The dehydrogenase (DHA) was measured by the procedure described by Casida Jr et al. (1964) and expressed as μg of triphenyl formazan released per g soil per day. The substrate-induced respiration (SIR) was measured the rate of respiration in the soil after glucose was amended in it and expressed as μg of $CO_2$ released/g soil/h (Enwall et al., 2007).

### 2.3. Data analysis

The relation between soil variables influenced by long-term nutrient management adoptions was evaluated by Pearson correlation analysis (Pearson, 1895) and simple linear regression (Freedman, 2009) using SPSS (SPSS Statistics for Windows, Version 20.0. Armonk, NY: IBM Corp). The scoring function for each assessed variables of soil was developed by SPSS 20.0. For this, the data were transformed into rank scores (rank case function of SPSS) and scoring percentile was calculated using the following formulae:

$$Percentile\ score = \frac{Ranking\ score\ of\ the\ variable - 0.05}{Number\ of\ samples} \times 100$$





In order to assess the relativeness of assessed soil variables and their cumulative contribution to the variability
among the treatments, principal component analysis (PCA) (Wold et al., 1987) was performed on the data using
XLSTAT (Version 2010.5.05, Addinsoft, USA).

*2.4. Estimating soil biological quality index (SBQI)*

2.4.1. Simple additive methods (SBQI-1 and SBQI-2)

In the simple additive method, the assessed soil parameters were given threshold values based on the
available literature and previous experiences. The threshold values of each parameter were further scored as
unitless soil index scores (SIS) (Supplementary Table 1). From these score values of the parameters, the soil
biological quality index (SBQI), unitless scoring value scaled to 1-10, was calculated using the formula as follows
(Amacher et al., 2007):

$$SBQI - 1 = \frac{\sum_{i=0}^{n} SIS}{S} \times 10$$

Where, SIS represents the score value of individual attributes; S represents the sum of maximum SIS (=24).
In SBQI-2, the index computed was normalized using the maximum and minimum values the dataset
(Amacher et al., 2007). The formula for this method is as follows:

$$Scaled\ SBQI = (\sum SIS - SIS_{min})/(SIS_{max} - SIS_{min})$$

$$SBQI - 2 = \frac{Scaled\ SBQI}{S} \times 10$$

Where, ΣSIS refers sum of all soil index scores and SISmin and SISmax are minimum and maximum values
of SIS of the dataset. S represents the sum of maximum SIS (=24)

2.4.2. Weighed additive method (SBQI-3)

For this, the data were transformed into rank scores (rank case function of SPSS) and scoring percentile was
calculated in SPSS. The scoring percentiles were summed and scaled to 10 (Moebius-Clune et al., 2016).  Further,
the index values were normalized using the minimum and maximum SBQI values of the dataset. The formulae for
the SBQI-3 calculation are as follows:

$$\sum SBQI = \frac{\sum Percentile\ score\ of\ individual\ attributes}{MP} \times 10$$

ΣSBQI represents the sum of SBQI derived from percentile scores, whereas MP represents the sum of the
maximum percentile score (=600).

$$SBQI - 3 = (\sum SBQI - SBQI_{min}) / (SBQI_{max} - SBQI_{min})$$

Where, ΣSBQI refers sum values from the above formula and SBQImin and SBQImax are minimum and
maximum values of SBQI of the dataset.





### 2.4.3. PCA based SBQI (SBQI-4)

The principal component analysis of all the six biological parameters pertaining to four soil samples of three locations was performed as described elsewhere. From the outcome of PCA, the SBQI was calculated (Andrews et al., 2002;Mandal et al., 2011;Masto et al., 2008). This SBQI used the percent contribution of individual variability to calculate the over-all soil biological quality of the soil. The formulae adopted to calculate SBQI-4 are as follows:

$$Cumulative\ variability\ (\%) = PC1\ variability + PC2\ variability$$

$$Individual\ variability\ contribution\ (VC) = \frac{\%\ contribution\ by\ the\ biological\ indicator}{Variability\ of\ the\ corresponding\ PC}$$

$$SBQI - 4 = \frac{\sum(Observed\ value\ \times VC)}{Cumulative\ variability}$$

### 2.4.4. Quadrant-plot based SBQI (SBQI-5)

As a soil variable is not independently acting and it is a dependent of several other variables or under the influence of other variables, the relativeness of the two closely-associated variables (Example SOC and MBC) is used to measure the soil biological quality. This method is adopted for the variables that are well-correlated to each other. Six significantly correlated (P <0.001) variable pairs and their $R^2$ values, means were used for the scoring (Supplementary Table 2). The paired variables were plotted in a scatter plot using variable-1 in x-axis and variable-2 in the y-axis. The scatter plot was converted into four quadrants by scaling the mean values of the corresponding
variables in their axes. The right-handed upper quadrant represents 'high' for both variables are scaled to 4, as both the variables above the means. The right-handed lower quadrant represents 'high for variable-a and low for variable-b' is scaled to 3. Likewise, left-handed upper quadrant scored for 2 and the left-handed lower quadrant which represents 'low' for both the variables had the value of 1. All the six-pairs (SOC/MBC, SOC/SLC, SOC/SIR, MBC/SPI, MBC/DHA, MBC/SIR) were scored using this method and SBQI was calculated as follows:

$$SBQI - 5 = \sum(Paired\ variable\ score\ \times regression\ coefficient)$$

### 2.5. Validation of SBQIs in farmers' field

To validate the SBQI methods, the soils collected from farmers' fields were assessed the soil biological indicators as described in previous chapter and the biological quality indices were calculated using the five methods as described earlier. The details of those soil samples were presented as Supplementary Table 3. All the
five SBQIs measured for long-term nutrient management adopted soils and farmers' soil were compared through Pearson correlation as described earlier in order to understand the effectiveness and relation of each other.

## 3. Results

### 3.1. Statistical scrutiny of soil biological attributes for developing SBQI

The histogram of measured values (x-axis) of each variable and its frequency (y-axis) with a distribution curve
or bell curve showed that the data observed were normally distributed. The mean ± SD for the observed





parameters viz., 7.29 ± 2.46 (SOC), 382.51 ± 199.61 (MBC), 480.30 ± 234.17 (SLC), 5.46 ± 0.84 (SPI), 11.51 ± 9.54 (DHA) and 3.20 ± 0.56 (SIR) were well-fit in the curve (Fig. 1). Among the six variables, the histogram of DHA (Fig. 1E) was skewed, while those others showed normal.

In correlation analysis, SOC had a significant correlation with other five biological variables, while MBC, SLC,
DHA, and SIR had a significant correlation with other variables except for SPI (Table 2). Similarly, SOC as an independent variable with others as the dependent variables, the linear regression coefficient ($R^2$) showed significance (Table 3). All the dependent variables (MBC, SLC, SPI, DHA, SIR) showed significant $R^2$ ($P<0.000$). However, SPI had the lowest $R^2$ (0.237), while the SLC had highest $R^2$ (0.417). Likewise, SPI had lowest but significant linear regression coefficient (0.089) with MBC, while with others had high $R^2$ values. SPI with other
variables such as SLC, DHA, and SIR had insignificant $R^2$.

The scatter plot with the interpolation curve between the actual values (x-axis) and the percentile scores (y-axis) had a similar trend and relation for all the assessed biological attributes (Fig. 2). The mean + SD of actual value had 79 to 81 percentile (Fig. 2A to 2F). Hence, all the six variables used in the present study fall under 'more is better' category, which implies that improving these variables will reflect the soil health.

The PCA-biplot representing the PC1 and PC2 of assessed variables and soil samples was presented in Fig. 3. PC1 had a variability of 75.21% and PC2 added 20.48% with a cumulative variability of 95.68%, which were due to six biological variables. All the soil parameters significantly contributed to the cumulative variability of PCs. Among the soil samples, OM and INM samples of Coimbatore and Madurai, which recorded highest and positively influenced due to the nutrient managements positioned in the right-hand top quadrant, while the control samples,
negatively impacted by the observed variables positioned in the left-hand bottom quadrant. The control soil of Madurai, which is at par with IC, OM and INM of Madurai and higher than Killikulam also positioned in the positive quadrant. All the variables except SPI significantly contributed to the PC1 (>0.80 loading value), while SPI had significant loading value to PC2. With reference to the contribution of individual soil variables to the total variability of the PC1 (75.21 %), MBC had 21.01%, SIR 19.88%, SLC 19.22 %, SIR 19.88%, and SOC had 18.64%
contributions. SPI had 64.75% contribution to the PC2 variability (20.48%) (Supplementary Table 4).

*3.2. SBQIs of long-term nutrient management-adopted soils*

The SBQIs of four long-term nutritionally managed soils were computed as a 10-scale unitless index using six biological attributes (Table 4). The sample-wise SBQIs calculated were presented as spread sheet (Supplementary file XLS). The SBQI-1 calculated using the threshold values of each biological attributes were ranged between 3.43
and 7.31 for the tested soil samples. Among the four nutrient managements, OM and INM had highest SBQI values (5.93 and 6.62 for Coimbatore; 7.04 and 7.31 for Madurai; 4.49 and 5.05 for Kovilpatti respectively). The wetland soil (Madurai) recorded the highest index followed by irrigated garden land (Coimbatore) and least in dryland (Kovilpatti). The least index values (between 3.0 and 4.0) were recorded in unfertilized control and IC soils. Overall, the SBQI-1 significantly discriminated the soils based on the soil index scales used by threshold index of respective
soil biological variables. SBQI-2 was derived from SBQI-1 after scaling it with minimum and maximum values. Hence, the SBQI-2 values were lower than the SBQI-1, without any change in the trends due to either treatments or centres (Table 4).

The SBQI-3 was calculated based on the scoring functions (percentile) of each assessed biological variable. The calculated soil biological quality index for the four different nutrient management enforced soils collected from
three different soil types (locations) showed a significant difference due to nutrient management as well as due to locations. In this method also, the highest biological index was recorded in the soils of Madurai (wetland soil) followed by Coimbatore (irrigated garden land soil) and least in Killikulam (dryland soils). Among the soils tested,



INM from Madurai recorded the highest SBQI of 8.39, followed by OM (Madurai) (7.59), while IC and control of Madurai recorded the quality index of 6.90 and 5.57, respectively. The Coimbatore (Alfisol) soils had SBQ index of
7.13 (INM), 6.25 (OM), 3.43 (IC) and 2.77 (Control), whereas the Kovilpatti soils recorded the lowest SBQI values. INM recorded 4.24, which is lower than Control soil of Madurai, OM with 3.42; IC with 2.57 and Control had 1.73. However, like the other two methods (SBQI-1 and SBQI-2), the resolution to discriminate the soils based on the biological properties due to long-term nutrient management is high for this method also.

From the PCA, the % contribution of each variable to the PCs (PC1 with SOC, MBC, SLC, DHA, and SIR; PC2
with SPI) was used to compute the SBQI-4. The actual values were weighed based on their % contribution in PCA to the total cumulative variability. As depicted from other SBQI methods, in this method also, the soils followed the same trends of SBQI values. The highest SBQI was recorded in INM (Madurai) with 6.59 followed by OM (Madurai) 6.05. Within Coimbatore centre, INM recorded the highest index of 5.22 followed by OM (5.89), IC (3.22) and control (3.24). The same trend was noticed for other centres also. In SBQI-5, the relation of two variables and
their measured values were used for computing the quality index. The paired variables were plotted in a scatter plot and the mean of both the variables was used to form quadrants of the plot (Figure 4). The samples positioned in the quadrants were scored (scaled from 1 to 4) and the score values were weighed with the regression coefficient ($R^2$) and scaled to 10. Such calculated SBQI-5 values for the long-term nutrient management enforced soils were the lowest among the five different methods. The Madurai soil (wetland) recorded a score value of 4.79 to 6.79,
which are relatively higher than Coimbatore (irrigated garden land soil) (2.14 to 6.43) and Kovilpatti (dryland) (1.94 to 3.95). With reference to the nutrient management effects, OM ≈ INM > IC > Control was the trend followed in three different soil types.

### 3.3. SBQIs of farmers' soils

All the five SBQI procedures scored the biological quality of the farmers' soil with the uniform trend among
them (Table 5). Irrespective of the soils, SBQI-1 had a high level of scaling (example 3.33 for sample A) followed by SBQI-2 (2.89), SBQI-5 (2.02), while SBQI-3 and SBQI-4 recorded 1.59 and 1.69, respectively. All the farmers' soils got lower SBQI scores (no soil with >6.0) compared to the SBQIs of long-term OM and INM soils of permanent manurial experimental soils. When the SBQI values of permanent manurial trial soils and farmers' field soils were pooled and assessed their relativeness, all the SBQI methods showed a significant positive correlation to each other
(Table 6).

## 4. Discussion

In the present work, we have developed a unitless soil biological quality index to scale the biological properties of the soil, in order to monitor the soil health. We have chosen six biological indicators viz., soil organic carbon, soil microbial biomass, soil labile carbon, soil protein index, dehydrogenase activity and substrate-induced
respiration, whose role in soil functioning are already well-documented. We measured these six variables from four different distinct soil samples that are under enduring influence of nutrient managements (control, inorganic fertilizer-applied, organic manure amended and integrated nutrient management adopted). Such long-term nutrient managements are being adopted by three different soils (semi-arid Alfisol – irrigated; semi-arid sub-tropical Alfisol-wetland; arid-Vertisol – dryland) with three different cropping sequences (maize-sunflower; rice-
rice; cotton-bajra, respectively). Hence, we assume that the data obtained from these three systems can be normalized and the impact of nutrient management to these soil biological attributes could be used to scale the SBQI so that the index can be applied to any range of soils of this region. With this background, the SBQI was



computed using these six biological indicators. Based on the literature and our previous works (Balachandar et al., 2016;Balachandar et al., 2014;Chinnadurai et al., 2013;Chinnadurai et al., 2014;Preethi et al., 2012;Tamilselvi et al.,
2015), it is obvious that these biological variables were significantly altered by the nutrient management adoptions. All these bio-indicators were reported highest in OM and INM, whereas the IC and control recorded on par values or sometimes IC was higher than control. Hence, the scale developed using these six variables should discriminate the OM, INM, IC and control to each other. We also assume that by comparing those SBQI values of long-term experimental plots to the farmer's soils, it may be possible to predict the biological quality of the soil. This approach
was already successfully used to compute the soil quality index (including physical, chemical and biological attributes) by Cornell University, USA as Cornell Soil Health Assessment (Moebius-Clune et al., 2016) and Soil Assessment and Management Framework by Soil Quality Institute (Andrews and Carroll, 2001;Wienhold et al., 2004;Wienhold et al., 2009).

Simple additive (SBQI-1) and scaled additive method (SBQI-2) used in the present investigation are the simple
aggregation of soil quality indicators (all the six of the present). Based on the literature and experts' opinions, each attribute is ranged into four scales (high -4; medium – 3; low – 2; very low – 1) and those are referred as 'soil index scales'. In SBQI-1, these scales were added and transformed to 1-10 scale, whereas in SBQI-2, these scale values were normalized using maximum and minimum score values. Compare to SBQI-1, SBQI-2 showed relative low SBQI. These simple additive methods performed well for the present soil ecosystems and discriminated the soils
based on their biological attributes as impact by the nutrient management adopted. In all the three locations, INM had high scores followed by OM, while IC and control had low index values. The consistent results obtained from all the three centres showed the efficiency of these two methods. Among the two, SBQI-2 would be more powerful than SBQI-1, as it normalizes the data based on the values of the data-set, which increased the resolution of the scoring giving weight to the localization of data. As pointed out by Mukherjee and Lal (2014), this method is
relatively simple, quick and user-friendly.

The SBQI-3 is based on the scoring functioning of assessed variables. It is an advanced way of calculating SQI, establishing standard non-linear scoring functions, which typically have shapes for 'more is better', 'optimum range', 'less is better' and 'undesirable range'. The scores are relative to the measured values of the respective region and transformed the values between 0 to 1, where 0 being poorest and score of 1 the best (Andrews et al.,
2004;Moebius-Clune et al., 2016). In the present work, all the measured values of six biological variables were scored for their percentile and non-linear scores obtained grouped them as 'more the better' shaped curved (Andrews et al., 2004;Moebius-Clune et al., 2016). As suggested by Moebius-Clune et al. (Moebius-Clune et al., 2016), mean + 1 SD was used to score the variables and all the six variables had 78-81% scoring functions, suggest that more than 70% of the samples fall within this range. Hence, these biological attributes could be the significant
contributors to the SBQI. If the values are less than 40%, the reliability of using the variable is questionable. In addition, to obtain the cumulative single index value, the scoring function percentiles of each variable were added, summed and normalized to scale between 1 to 10. The major assumption made in this method is that summing the scoring values (percentiles) of each variable rather than actual values or their soil index scales (as in case of SBQI-1 and SBQI-2) can provide more accurate score values among the samples tested. The scoring functions and the
plots are in accordance with the Cornell Soil Health Assessment (Moebius-Clune et al., 2016). The SBQI scored based on this method also had high discriminative power on the samples obtained from permanent manurial experiments of three different crops and soils. Among the three locations, dryland soils had the lowest SBQI in this method, while the wetland soils had the highest values. In all the three systems, INM>OM>IC>control is the trend followed for the SBQI-3 values.





The PCA-based calculation is the most popular method among the researchers worldwide, across the soil types and land use management to score the SQI (Bünemann et al., 2018).This method integrated the measured variables into PCs and used for scale them to SQI. In the present investigation, we have adopted the same method with slight modification. From the PCA factor loading, each variable's contribution to the corresponding PC was used to weigh the actual measured values and these weighed values were further summed and scaled to 1-10.

Unlike previous investigators (Biswas et al., 2017;Mukherjee and Lal, 2014;Schmidt et al., 2018), we have not picked the single variable for each PC, rather all the factor loadings of six biological attributes were used to scale the SBQI. This method also significantly discriminated the soils that are under the influence of long-term nutrient management adoptions under three different soil and crop types. Compare to all the above methods, this method is a more statistical approach and gives more stress to discriminate than other methods. This method was also

successfully used to measure the SQI and can able to predict the yield of a particular system (Mukherjee and Lal, 2014) and relating the soil functioning (Vasu et al., 2016).

The fifth method adopted to measure the SBQI from the available data is unique and uses the relatedness of two potential variables. The possible combinations of the variable pairs used are SOC/MBC, SOC/SLC, SOC/SIR, MBC/SPI, MBC/DHA, and MBC/SIR assuming that SOC and MBC are the major driving forces of the soil biology,

while the other four variables are relating to them to the functioning. The scatter plots of each pair of variables were divided into four quadrants using the mean of each corresponding variable. The assumption made here is that any sample having more than local-average is considered as 'high' and less than that is 'low'. Thus, relatedness of the two variables can divide the scatter plot into four quadrants, as 'high/high', 'high/low', 'low/high' and 'low/low'.  Based on the position of the samples in the four quadrants, score values were given ('high/high' - 4,

'high/low'-3, 'low/high' -2 and 'low/low'-1) and these score values were used to compute the SBQI. This method measured the four soils with least SBQIs, suggest that more pressure has been made to show the variability. This method adopts the less statistical and more biological approach to score the SBQI, unlike SBQI-3 and SBQI-4, which are more statistical and less biological.  Though the method is relatively complicated to compute the SBQI, more inference and the better understanding of soil biological variables can be obtained. For example, high SOC/high

MBC means the samples are sufficient with SOC and MBC, need to maintain them using organic amendments; high SOC/low MBC means the SOC may be recalcitrant or microbial inhibitors/heavy metals/pollutants may be present; need proper reclamation; low SOC/high MBC means the soil needs continuous organic amendments to proliferate the microbial growth; low SOC/low MBC means the soil biological quality is very poor; needs remedy to improve them. Like this, quadrant-based analyses can identify the 'soil biological constraints' more sensitively

than those methods.  However, more validation and reproducibility for different soil types are needed for this method before going for adoptions.

To validate the SBQIs developed during the present investigation, twenty-five farmers' field in and around Coimbatore and Nilgiris districts of Tamil Nadu state, India have assessed and SBQIs were computed by all the five models as detailed earlier. All the five SBQIs were in the same trend in the farmer's field. Compare to LTF

soils, the farmers' soils are low in SOC, MBC and all the measured attributes hence recorded lower SBQIs. In these soils also, SBQI-1 and SBQI-2 had relatively higher values followed by SBQI-3 and SBQI-4, while least was observed in SBQI-5. Soil from Ooty (Nilgiris) had relatively high SBQI scores compared to other samples. This was mainly due to the temperate climate and high SOC of those soils. Our SBQI results are as comparable to the three methods validated by Mukherjee and Lal (2014). The SBQI values measured in the farmers' fields identified following

constraints in the soil biological functioning: Most of the farm soils are with low SBQI values (< 4.0) and are in 'low SOC/low MBC', 'low MBC/low DHA' and 'low MBC/low SPI' category. The soil biological activities responsible for nutrient transformation, organic decomposition, carbon assimilation are low in these soils. The microbes are



under stress condition due to low resources available for them. The natural resources (soil nutrients) had an insignificant role to provide nutrient to the crops. Hence, continuous exogenous nutrient supply is needed for the
crops, failing which will impact the productivity. As the soil microbial and biochemical processes are of low magnitude, the resilience of the crops to any adverse conditions like drought, flood or high temperature is questionable. As the poor soil management continues, these soils may deter their quality which may reflect the productivity of subsequent crops.

## 5. Conclusions

In the present work, we have investigated the four-different nutrient management on soil biological attributes and the difference between them was used to scale a single unitless quantitative measure as SBQI. Five different models were proposed to compute the SBQI and each method discriminated the four soil samples accurately and we could not find any difference among them. However, each method has its own advantages and limitations. All the five methods gave the same results in the farmers' field and all the SBQI had a significant positive correlation
to each other. Among the five SBQI models tested, SBQI-5 would be an appropriate method, as it is with less statistics and more biological approach. This method also identifies the constraints of the soil biology better than the other four methods.

### Data availability

The data that support the findings of this study are available by request from the corresponding author
(D Balachandar).

### Author contributions

DB designed the experimental setup. SA and CC did the soil sampling and led the lab analysis procedure. DB also did the statistics, prepared the manuscript with valuable contributions of the two co-authors SA and CC and undertook the revisions during the review process.

### Competing interests

The authors declare that they have no conflict of interest.

### Acknowledgments

The financial support from Indian Council on Agricultural Research, New Delhi, India through All India Network Project (AINP) on Soil Biodiversity and Biofertilizers to conduct these experiments is acknowledged. Dr. K. Arulmozhiselvan, Professor, Department of Soil Science and Agricultural Chemistry, Tamil Nadu Agricultural University, Coimbatore, India is acknowledged for his help and support to collect the soil samples from the permanent manurial experimental fields.

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





Table 1. Details of the permanent manurial trails used for the present study

| Details | Coimbatore | Madurai | Kovilpatti |
|---|---|---|---|
| Centre | TNAU, Coimbatore | AC & RI, Madurai | ARS, Kovilpatti |
| Geographical coordinates | 11°N, 77°E | 9.97°N, 78°E | 09.12°N, 77.53°E |
| Altitude | 426 m | 147 m | 106 m |
| Max and Min temperature | 34.2°C and 20°C | 32°C and 23°C | 36°C and 29°C |
| Annual rainfall | 670 mm | 1100 mm | 730 mm |
| Climate type | semi-arid sub-tropical | arid sub-tropical | semi-arid tropic |
| Soil type | red sandy loam | sandy clay loam | Clayey |
| Soil classification | Typic Haplustalfs | Typic Haplustalfs | Typic Chromustert |
| Soil order | Alfisol | Alfisol | Vertisol |
| Year of establishment | 1909 | 1975 | 1982 |
| Test crop | Maize – Sunflower | Rice – Rice | Cotton |
| Cropping method | Irrigated | Wetland | Dryland |
| Variables | Nutrient management | Nutrient management | Nutrient management |





Table 2. Correlation coefficient (Pearson, n-1) of the observed variables from long-term nutrient management soils

| Variables | SOC | MBC | SLC | SPI | DHA | SIR |
|---|---|---|---|---|---|---|
| SOC | **1.00** | | | | | |
| MBC | **0.93** | **1.00** | | | | |
| SLC | **0.74** | **0.85** | **1.00** | | | |
| SPI | **0.68** | 0.51 | 0.10 | **1.00** | | |
| DHA | **0.65** | **0.81** | **0.95** | 0.05 | **1.00** | |
| SIR | **0.80** | **0.89** | **0.93** | 0.25 | **0.85** | **1.00** |

SOC – Soil organic carbon; MBC – Microbial biomass carbon; SLC – Soil labile carbon; SPI – Soil protein index; DHA –
Dehydrogenase; SIR – Substrate induced respiration. Values in bold are different from 0 with a significance level p=0.05.




Table 3. Regression analysis of soil variables assessed for long-term nutrient management adopted soils

| Independent variable | Dependent variable | $R^2$ | F | P |
|---|---|---|---|---|
| SOC | MBC | 0.237 | 32.95 | 0.000 |
| SOC | SLC | 0.417 | 75.77 | 0.000 |
| SOC | SPI | 0.283 | 41.79 | 0.000 |
| SOC | DHA | 0.329 | 51.97 | 0.000 |
| SOC | SIR | 0.409 | 73.34 | 0.000 |
| MBC | SLC | 0.256 | 36.42 | 0.000 |
| MBC | SPI | 0.089 | 10.36 | 0.002 |
| MBC | DHA | 0.259 | 37.03 | 0.000 |
| MBC | SIR | 0.337 | 53.90 | 0.000 |
| SLC | SPI | 0.006 | 0.62 | 0.435 |
| SLC | DHA | 0.834 | 534.10 | 0.000 |
| SLC | SIR | 0.662 | 207.80 | 0.000 |
| SPI | DHA | 0.003 | 0.324 | 0.571 |
| SPI | SIR | 0.023 | 2.53 | 0.115 |
| DHA | SIR | 0.604 | 161.68 | 0.000 |

SOC – Soil organic carbon; MBC – Microbial biomass carbon; SLC – Soil labile carbon; SPI – Soil protein index; DHA – Dehydrogenase; SIR – Substrate induced respiration. $R^2$ – regression coefficient (linear); F – F test; P - p value.





Table 4. Soil biological quality index of long-term nutrient management adopted soils of three different centres
assessed by five different methods (SBQI-1 to 5)

| Centre | Treatments | SBQI-1 | SBQI-2 | SBQI-3 | SBQI-4 | SBQI-5 |
|---|---|---|---|---|---|---|
| Coimbatore | Control | 3.66 (± 0.40) | 2.62 (± 0.40) | 2.77 (± 0.55) | 2.34 (± 1.41) | 2.14 (± 0.74) |
| | IC | 4.07 (±0.68) | 3.03 (± 0.68) | 3.43 (± 1.19) | 3.22 (± 1.99) | 2.86 (± 1.03) |
| | OM | 5.93 (± 0.46) | 4.88 (± 0.46) | 6.25 (± 0.53) | 4.89 (± 1.89) | 5.32 (± 0.86) |
| | INM | 6.62 (± 0.25) | 5.58 (± 0.25) | 7.13 (± 0.42) | 5.22 (± 0.86) | 6.43 (± 0.59) |
| Madurai | Control | 6.06 (± 0.37) | 5.02 (± 0.37) | 5.57 (± 0.61) | 5.02 (± 1.23) | 4.79 (± 1.16) |
| | IC | 6.53 (± 0.21) | 5.49 (± 0.21) | 6.90 (± 0.43) | 5.30 (± 1.43) | 5.74 (± 0.75) |
| | OM | 7.04 (± 0.39) | 6.00 (± 0.39) | 7.59 (± 0.53) | 6.05 (± 1.25) | 6.80 (± 0.34) |
| | INM | 7.31 (± 0.42) | 6.27 (± 0.42) | 8.39 (± 0.55) | 6.59 (± 1.29) | 6.79 (± 0.54) |
| Kovilpatti | Control | 3.43 (± 0.28) | 2.38 (± 0.28) | 1.73 (± 0.34) | 2.24 (± 1.16) | 1.94 (± 0.54) |
| | IC | 3.89 (± 0.36) | 2.85 (± 0.36) | 2.57 (± 0.55) | 2.47 (± 1.12) | 2.00 (± 0.53) |
| | OM | 4.49 (± 0.50) | 3.45 (± 0.50) | 3.42 (± 0.78) | 3.09 (± 1.31) | 2.92 (± 1.15) |
| | INM | 5.05 (± 0.67) | 4.01 (± 0.67) | 4.24 (± 1.21) | 4.02 (± 1.47) | 3.95 (± 1.26) |

Values are mean (± SD) of three replicates. Control - Unfertilized control soil; IC - Inorganic chemical fertilized soil; OM - Organically managed soil; INM - Integrated nutrient management enforced soil; SBQI1 - SBQI5 refer the unitless 10-scaled soil biological quality index computed using six soil biological variables.





Table 5. SBQI values of farmers' soils measured by five different methods

| Farmers' field | SBQI-1 | SBQI-2 | SBQI-3 | SBQI-4 | SBQI-5 |
|:---:|:---:|:---:|:---:|:---:|:---:|
| A | 3.33 | 2.89 | 1.59 | 1.69 | 2.02 |
| B | 3.75 | 3.31 | 2.06 | 2.22 | 1.76 |
| C | 4.17 | 3.73 | 1.72 | 1.80 | 1.76 |
| D | 3.33 | 2.89 | 2.05 | 2.18 | 1.76 |
| E | 4.58 | 4.14 | 2.33 | 2.46 | 2.02 |
| F | 4.17 | 3.73 | 2.40 | 2.56 | 2.12 |
| G | 5.00 | 4.56 | 2.91 | 3.12 | 2.12 |
| H | 3.33 | 2.89 | 1.22 | 1.25 | 1.86 |
| I | 5.42 | 4.98 | 2.45 | 2.60 | 2.12 |
| J | 3.75 | 3.31 | 1.81 | 1.90 | 2.02 |
| K | 3.33 | 2.89 | 1.45 | 1.50 | 1.76 |
| L | 3.75 | 3.31 | 1.68 | 1.77 | 2.02 |
| M | 5.00 | 4.56 | 2.28 | 2.37 | 2.12 |
| N | 5.42 | 4.98 | 2.61 | 2.72 | 2.12 |
| O | 4.58 | 4.14 | 2.95 | 3.14 | 2.12 |
| P | 3.75 | 3.31 | 1.55 | 1.64 | 2.02 |
| Q | 3.75 | 3.31 | 2.07 | 2.20 | 2.02 |
| R | 4.58 | 4.14 | 1.92 | 2.01 | 2.12 |
| S | 4.17 | 3.73 | 1.90 | 2.03 | 2.02 |
| T | 4.17 | 3.73 | 1.47 | 1.52 | 2.02 |
| U | 4.58 | 4.14 | 3.88 | 4.20 | 3.49 |
| V | 5.42 | 4.98 | 2.64 | 2.78 | 2.86 |
| W | 4.58 | 4.14 | 4.36 | 4.67 | 3.81 |
| X | 5.42 | 4.98 | 2.20 | 2.32 | 1.86 |
| Y | 4.58 | 4.14 | 4.98 | 5.36 | 3.91 |

SBQI-1 to SBQI-5 represent the unitless 10-scaled values of soil biological quality index calculated for the farmers' field soil using different methods as described earlier. Details of farmers' field soils are provided in Supplementary Table S3.





Table 6. Correlation coefficient (Pearson (n-1)) relating five different methods used to measure the soil biological quality index of long-term nutrient management adopted soils of three different agro-ecological zones and farmers' soils of Tamil Nadu

| SBQI methods | SBQI-1 | SBQI-2 | SBQI-3 | SBQI-4 | SBQI-5 |
|---|---|---|---|---|---|
| SBQI-1 | **1.00** | | | | |
| SBQI-2 | **0.97** | **1.00** | | | |
| SBQI-3 | **0.85** | **0.75** | **1.00** | | |
| SBQI-4 | **0.82** | **0.73** | **0.97** | **1.00** | |
| SBQI-5 | **0.84** | **0.73** | **0.98** | **0.94** | **1.00** |

Values in bold are different from 0 with a significance level alpha=0.0; SBQI-1 to SBQI-5 represent the unitless 10-scaled values of soil biological quality index calculated for the soil samples.





**Fig. 1.** Histogram and distribution curve (bell curve) of the observed soil biological variables from four different nutrient management plots of three different agro climatic zones of Tamil Nadu, India. A - Soil organic carbon; B - Microbial biomass carbon; C - Soil labile carbon; D - Soil protein index; E - Dehydrogenase activity; F - Substrate-induced respiration.






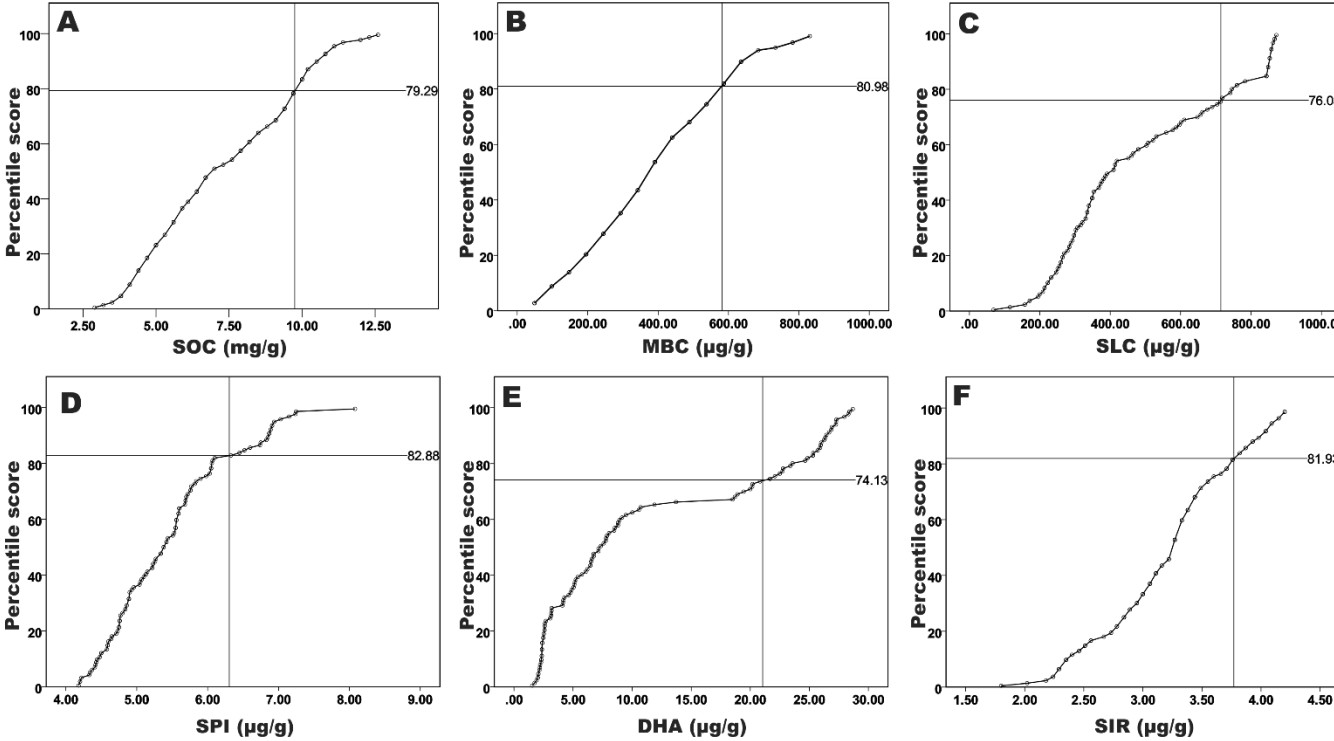

**Fig. 2.** Cumulative normal distribution for scoring the observed soil biological variables in four different nutrient management plots of three different agro climatic zones of Tamil Nadu, India. A - Soil organic carbon; B - Microbial biomass carbon; C - Soil labile carbon; D - Soil protein index; E - Dehydrogenase activity; F - Substrate- induced respiration. In the distribution curve, the mean + SD of measured values were intercepted and the scoring percentile for each variable was calculated and presented in the corresponding plot.







**Fig. 3.** Principal component analysis biplot showing the relation between the soil biological variables in four different nutrient management plots of three different agro climatic zones of Tamil Nadu, India. SOC - Soil organic carbon; MBC - Microbial biomass carbon; SLC - Soil labile carbon; SPI - Soil protein index; DHA - Dehydrogenase activity; SIR- Substrate-induced respiration. Control - Unfertilized control soil; IC - Inorganic chemical fertilized soil; OM - Organically managed soil; INM - Integrated nutrient management enforced soil; C - Coimbatore; M - Madurai; K - Kovilpatti. The % variance explained by each component (PC1and PC2) is given in parentheses in axes.







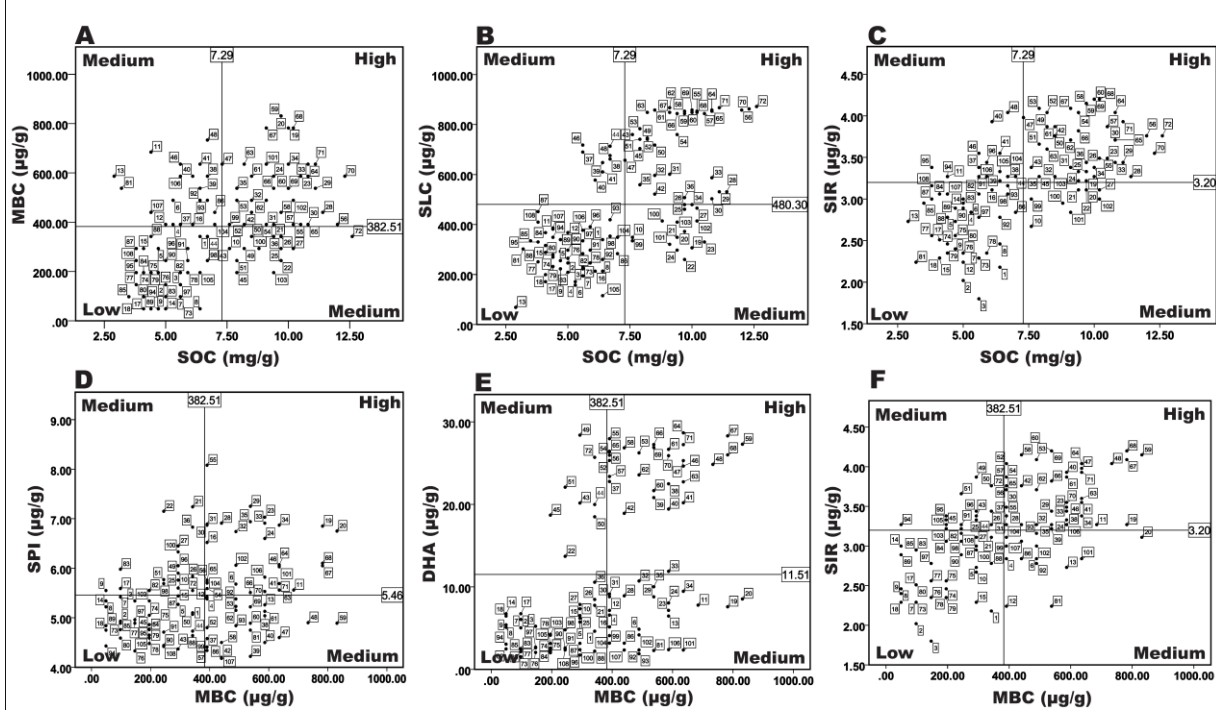

**Fig. 4.** Quadrant scatter plots showing the relatedness of the soil biological variables in four different nutrient management enforced soils of three different agro climatic zones of Tamil Nadu. Each scatter plot is divided into quadrants based on the mean of respective axis, which are indicated in the plot. Quadrant with 'High' represents both the variables are above the average; 'Medium' represents any one of the variables is below the average; 'Low' represent both the variables are below the average. Main variable is in x axis and secondary variable for it is in y axis. 1-108 represent the soil samples.