# Peer review of "Original Research article"

_SOIL, 2019_

## Referee Comment (RC1) · Anonymous Referee #1 · 20 Dec 2019

The topic presented in this manuscript is of high interest. A valid and general soil assessment is urgently required and there are many attempts to develop soil quality indices (SQI). This publication adds to the necessary discussion, although the suggested SQI (or biological SBQI) appears not to be the final solution. The authors investigated three different long-term fertilization experiments, each with four different fertilization systems. In total 12 soils were used to develop the scoring scheme that was subsequently tested with 25 farm soils. The parameters selected were 1) soil organic carbon, 2) microbial biomass carbon, 3) labile carbon, 4) protein index, 5) dehydrogenase activity and 6) substrate-induced respiration. Data were converted into SBQI, testing five different methods. In his fundamental publication, Nortcliff (2002) stated that any SQI must consider soil functions. Since soil functioning largely depends on

soil biota, it is agreed that the authors focus on the soil biological status. However, the parameters selected are all measures of quantities; the dehydrogenase activity and substrate-induced respiration are microbial activities, yet they are so widespread and unspecific that they mostly reflect microbial biomass. That can be seen also in this study from the very close correlation of microbial biomass carbon with dehydrogenase activity and substrate-induced respiration (Tab. 2). All these parameters again very much depend on the content of soil organic carbon. Correspondingly, the loading of all these parameters on the first principal component in principal component analysis (PCA) is very similar (Tab. S4). Anyhow, the authors are able to clearly differentiate the 12 samples from long-term experiments by using PCA. Yet, it must be assumed that this differentiation is mostly due to the systematic differences in the soil samples, which result from the different treatment in the long-term experiments. The authors should test, in how far soil samples are separated solely by soil organic carbon and protein index (and vice versa, whether the other parameters are dispensable). The test of five different methods for deriving a SBQI from the data set nicely shows that it doesn't depend so much on the calculation method. This is shown by the results in Tab. 5 (and partly Tab. 6). Method 1 and 2 as well as 3 and 4 deliver content-related the same data (the regression coefficient is 1 and not 0.97 as written in Tab. 6). Even more relevant and worth to discuss is the interpretation of the derived SBQI. How do we know which value is natural, desirable, sustainable? The authors come up with a "the more the better" target, based on average values calculated from their data. Yet, is for example more respiration better than lower respiration? Here it seems to be a more promising approach to go for ratios such as the metabolic quotient, the microbial quotient or the carbon use efficiency of soil microbiota. The text contains a number of small language errors that have to be fixed by careful proof reading and editing. As was already indicated, the data in Table 6 must be checked and corrected. Normal distribution of data is claimed by the authors. However, the Fig. 1 shows clear deviations from normal distribution. Distributions are left-skewed for soil organic carbon and labile carbon, while dehydrogenase activities show a bimodal distribution. Nortcliff, S. (2002)

Standardisation of soil quality attributes. Agriculture, Ecosystems and Environment 88(2), 161-168.

---

## Referee Comment (RC2) · Anonymous Referee #2 · 24 Dec 2019

Writing in the manuscript is pretty good for non-native English speakers, but the manuscript needs English editing. I suggest the authors use an English editing service or ask for assistance from a native English speaking colleague.

Lack of adequate explanation of the study design and the need for improved English in places make it difficult to fully evaluate this manuscript. I have done my best below to provide the authors with feedback to improve the manuscript, but the two points mentioned above need to be dealt with and the manuscript will need to be reviewed again.

Line 33 – While I agree with the statement here about the lack of a common set of soil quality indicators, and I have a great deal of respect for Bouma's work, the paper cited here is nearly 20 years old. Many changes have occurred in the soil quality world over

the last 20 years. It seems that a more recent reference would be well advised here, given that you are using it to support a statement about the current status of the field. Something from the last 3 years or so would be much more appropriate.

Most of the references used in the introduction are more than 10 years old. This can leave one wondering if the topic of the paper is outdated. Can the author provide some more recent references for this section?

Lines 63-65 – 11 references really are not needed here to support this idea, these should be reduced, at least cut in half.

Lines 68-70, Again, 11 references are not needed.

Lines 74-81 – This section belongs in Materials and Methods.

Lines 75-76 – Do not need six refences here.

Line 86 (and other places as appropriate) – I suggest using "manure trials" rather than "manurial trials".

Line 90 – I would say that Table 1 gives the individual site characteristics. With the current wording "The details of long-term permanent manurial trials are described" I expected to see information on soil characteristics, such as organic matter content, various nutrient contents, infiltration rates, etc. The information given in Table 1 is completely appropriate and important, just mislabeled.

Table 1 – The authors are using US Soil Taxonomy, but the soil types given do not correspond to any formal classification level in Soil Taxonomy. Should this be soil texture? All three entries carry textural information, although the "red" in the "red sandy loam" entry is irrelevant to texture. Is this a local Indian classification? If so, this should be made clear with a footnote at the bottom of the table. The soil order entry is not needed in the table, that information is already provided in the soil classification entry. Is it possible to be any more specific than "nutrient management" in the variables entry? What about nutrient management varied? And is this referring to different nutrient

management within each site, or between sites?

Lines 90-93 – More detail is desirable here. What were the application rates (or ranges) of inorganic fertilizers at each site? What were the N-equivalent manure applications (or ranges) at each site? Etc.

Lines 93-96 – We need more detail on the study design. Were the plots at each of the three research site, and if so, how many? Or was each research site treated as a single plot? How many soil samples were actually collected and analyzed, from each treatment and total for the study? What do you mean by "nine such replicates were maintained per soil"? Are you saying you broke the composite samples (from the 10 random cores) into 9 subsamples, or are you saying you collected 10 random cores from nine different sites at each of the study sites? Or maybe it is something else (I can come up with other possibilities). In short, the experimental design section needs to be expanded upon. I cannot picture the experimental design based on this description, and if readers cannot understand the experimental design nothing past this point matters, this manuscript is not publishable, because we cannot adequately evaluate the work and what the results actually mean.

Line 96 – Were samples stored on ice while in the field during collection?

Line 121 – The process used to establish the SBQ values is critical to evaluating this work, both for the reviewers right now and for future readers. Supplementary Table 1 is critical to this, and should therefore not be a supplementary table. The process of developing these threshold values needs to be clearly and completely described, therefore this table needs to be part of the manuscript.

Lines 145-146 – Where did the four soil samples from the three locations come from? I assume the three locations are the three study sites, but where do the four soil samples come from. See comment above about the need to clearly and completely describe the experimental design.

Lines 150-153 – Source(s) of these formulae?

Line 158 – Again, I think this table is too important to understanding the manuscript to be a supplemental table.

Lines 161-162 – Why is "high for variable a and low for variable b" scored 3 while "low for variable a and high for variable a" scored 2?

Lines 166-171 – Where did the farmers' fields come from? How and why were they chosen? How do they compare to the three study sites (cropping systems, soils, climate, etc., see Table 1)? You cannot just randomly drop another set of variables into the middle of a scientific paper and not explain them.

Table 2 – Using a symbol such as an * would be a better way to indicate significance than bold values. Bold may not translate well if the paper gets copied in the future.

Table 3 – The p values (where appropriate) are <0.001, not 0.000. Same comment in Line 182.

Section 3.3 – Details of the farmers fields have not been provided, so we do not know what we are making comparisons to here and in Table 6. I see the reference in Table 6 to supplementary materials, but this is critical information. It needs to be part of the paper, not part of the supplementary materials.

Line 251 – I have already discussed the need to better explain the experimental design, but four distinct soil samples? Really? You have three study sites with four treatments at each study site, shouldn't you have, at a bare minimum, 12 samples?

Lines 254-255 – Now different cropping sequences are being brought in. I don't remember any previous mention of different cropping sequences. Explanation of the experimental design for this project is extremely confusing!

Lines 258-260 – Don't need six citations here.

Lines 260-262 – Are there works from other research groups (other than the authors)

[Figure]

that would indicate the results found here are reasonable? How about results from other environmental settings? SOIL is an international journal, to be publishable in SOIL this work needs to be tied into the bigger picture, the authors need to show why there should be international interest and not just interest within India.

Lines 269-273 – This is just a repeat of what has already been said in the Materials and Methods. Not needed here.

All the SQIs were found capable of separating the soils and treatments studied, and each has its own strengths and weaknesses. So, which of the SQI approaches do the authors recommend based on this work? Did on perform better than the others based on some objective method, was one better than the others because of its simplicity, or independence from "expert" (subjective) opinion, etc.? You get at this in the Conclusion, but seem to indicate SQI5 is the best after saying in Lines 330-331 that SQI5 needs more investigation/development.

Starting on Line 332 – As previously mentioned, more information on the farmers' fields, how they were chosen, what their soils and management are, etc. needs to be supplied earlier in the manuscript.

---

## Author Comment (AC1) · 11 Jan 2020

**Reply to the reviewers**

MS Title: Development of soil biological quality index for soils of semi-arid tropics
Author(s): Selvaraj Aravindh et al.
**MS No.: soil-2019-60**

All the points mentioned in the interactive comment of **Anonymous referee #1** were carefully considered and suitable reply for each point in given in red font. All the replies were compiled as Single PDF file and attached with this reply.

**Anonymous referee #1**

The topic presented in this manuscript is of high interest. A valid and general soil assessment is urgently required and there are many attempts to develop soil quality indices (SQI). This publication adds to the necessary discussion, although the suggested SQI (or biological SBQI) appears not to be the final solution.

Reply: As pointed out by referee, this is not the final solution for measuring soil quality. It is an attempt to narrow down the issues to measure the soil quality (especially agricultural soils) through identifying potential indicators. Here, we should substantiate that these six variables were chosen based on their consistent performance, relatively quick and simple for assessment and informative.

The authors investigated three different long-term fertilization experiments, each with four different fertilization systems. In total 12 soils were used to develop the scoring scheme that was subsequently tested with 25 farm soils. The parameters selected were 1) soil organic carbon, 2) microbial biomass carbon, 3) labile carbon, 4) protein index, 5) dehydrogenase activity and 6) substrate-induced respiration. Data were converted into SBQI, testing five different methods. In his fundamental publication, Nortcliff (2002) stated that any SQI must consider soil functions. Since soil functioning largely depends on soil biota, it is agreed that the authors focus on the soil biological status. However, the parameters selected are all measures of quantities; the dehydrogenase activity and substrate-induced respiration are microbial activities, yet they are so widespread and unspecific that they mostly reflect microbial biomass. That can be seen also in this study from the very close correlation of microbial biomass carbon with dehydrogenase activity and substrate-induced respiration (Tab. 2). All these parameters again very much depend on the content of soil organic carbon. Correspondingly, the loading of all these parameters on the first principal component in principal component analysis (PCA) is very similar (Tab. S4). Anyhow, the authors are able to clearly differentiate the 12 samples from long-term experiments by using PCA. Yet, it must be assumed that this differentiation is mostly due to the systematic differences in the soil samples, which result from the different treatment in the long-term experiments.

Reply: As referee pointed out, there is no single soil parameter to measure the soil quality. Hence, we tried to use inter-related biological variables and their values to measure the soil functioning. We agree that all the five variables are dependent of soil organic carbon. However, the SOC alone cannot able to describe the soil functioning. Hence, we used microbial biomass carbon and dehydrogenase as the total amount of microbiome; soil protein index and labile carbon as indicator for available nutrients for the microorganisms; substrate induced respiration represents the microbial activity as influenced by these factors. All these variables are inter-related to each other, making their measures into soil biological quality index was quite easy and we

demonstrated that these values are constant for the soils under different nutrient management regimes.

The authors should test, in how far soil samples are separated solely by soil organic carbon and protein index (and vice versa, whether the other parameters are dispensable).

Reply: Before the cumulative approach as described in the manuscript, we have assessed the SBQI considering SOC as independent sole indicator representing soil quality. Likewise, for each variable, but the method did not show any apparent conclusion. Hence, we adopted minimum data set method, for which we have chosen these six inter-related variables and their threshold values to measure the SBQI.

The test of five different methods for deriving a SBQI from the data set nicely shows that it doesn't depend so much on the calculation method. This is shown by the results in Tab. 5 (and partly Tab. 6). Method 1 and 2 as well as 3 and 4 deliver content-related the same data (the regression coefficient is 1 and not 0.97 as written in Tab. 6).

Reply: As mentioned by referee, theoretically method 1 and method 2 as well as method 3 and method 4, which used same set of data should have regression coefficient of 1.0. However, we re-analyzed the data and found the typo-error. The statistical analysis gave 0.995 (due to decimal changes in the processed data) which will be corrected in Table 6 of revised MS.

Even more relevant and worth to discuss is the interpretation of the derived SBQI. How do we know which value is natural, desirable, sustainable? The authors come up with a "the more the better" target, based on average values calculated from their data. Yet, is for example more respiration better than lower respiration? Here it seems to be a more promising approach to go for ratios such as the metabolic quotient, the microbial quotient or the carbon use efficiency of soil microbiota.

Reply: During revision, we will include the suggestion about the 'natural, desirable and sustainable values' and improve the interpretation for 'more the better'.

The text contains a number of small language errors that have to be fixed by careful proof reading and editing.

Reply: We will thoroughly check for language errors.

As was already indicated, the data in Table 6 must be checked and corrected.

Reply: As pointed out by the referee, we have reanalyzed the data and found error in transferring values as table form, this will be corrected in the revised MS.

Normal distribution of data is claimed by the authors. However, the Fig. 1 shows clear deviations from normal distribution. Distributions are left-skewed for soil organic carbon and labile carbon, while dehydrogenase activities show a bimodal distribution.

Reply: The discussion of the above result will be changed as left-skewed and bimodal distributions in the revised MS.

---

## Author Comment (AC2) · 11 Jan 2020

**Reply to the reviewers**

MS Title: Development of soil biological quality index for soils of semi-arid tropics
Author(s): Selvaraj Aravindh et al.
**MS No.: soil-2019-60**

All the points mentioned in the interactive comment of **Anonymous referee #2** were carefully considered and suitable reply for each point in given (in red font). All the replies were compiled as Single PDF file and attached with this reply.

**Anonymous referee #2**
Writing in the manuscript is pretty good for non-native English speakers, but the manuscript needs English editing. I suggest the authors use an English editing service or ask for assistance from a native English speaking colleague. Lack of adequate explanation of the study design and the need for improved English in places make it difficult to fully evaluate this manuscript. I have done my best below to provide the authors with feedback to improve the manuscript, but the two points mentioned above need to be dealt with and the manuscript will need to be reviewed again.

Reply: As per the referee's suggestion, we will improve the English of the MS through native English speaker, so as to provide adequate explanation of the study design.

Line 33 – While I agree with the statement here about the lack of a common set of soil quality indicators, and I have a great deal of respect for Bouma's work, the paper cited here is nearly 20 years old. Many changes have occurred in the soil quality world over the last 20 years. It seems that a more recent reference would be well advised here, given that you are using it to support a statement about the current status of the field. Something from the last 3 years or so would be much more appropriate.

Reply: We will include recent references during revision.

Most of the references used in the introduction are more than 10 years old. This can leave one wondering if the topic of the paper is outdated. Can the author provide some more recent references for this section?

Reply: We will include recent references during revision.

Lines 63-65 – 11 references really are not needed here to support this idea, these should be reduced, at least cut in half.

Reply: We will reduce to 5 in the revised MS.

Lines 68-70, Again, 11 references are not needed.

Reply: We will reduce them in revised MS.

Lines 74-81 – This section belongs in Materials and Methods.

Reply: We will move to Materials and Methods during revision.

Lines 75-76 – Do not need six references here.
Reply: All the six references are our work which are essentially needed for describing the background.

Line 86 (and other places as appropriate) – I suggest using "manure trials" rather than "manurial trials".

Reply: As per the suggestion, we will use "manure trials"

Line 90 – I would say that Table 1 gives the individual site characteristics. With the current wording "The details of long-term permanent manurial trials are described" I expected to see information on soil characteristics, such as organic matter content, various nutrient contents, infiltration rates, etc. The information given in Table 1 is completely appropriate and important, just mislabeled.

Reply: We will do correct labelling of Table 1 in revised MS.

Table 1 – The authors are using US Soil Taxonomy, but the soil types given do not correspond to any formal classification level in Soil Taxonomy. Should this be soil texture? All three entries carry textural information, although the "red" in the "red sandy loam" entry is irrelevant to texture. Is this a local Indian classification? If so, this should be made clear with a footnote at the bottom of the table. The soil order entry is not needed in the table, that information is already provided in the soil classification entry. Is it possible to be any more specific than "nutrient management" in the variables entry? What about nutrient management varied? And is this referring to different nutrient management within each site, or between sites?
Lines 90-93 – More detail is desirable here. What were the application rates (or ranges) of inorganic fertilizers at each site? What were the N-equivalent manure applications (or ranges) at each site? Etc.

Reply: We will include all the corrections/suggestions as pointed out by referee.

Lines 93-96 – We need more detail on the study design. Were the plots at each of the three research site, and if so, how many? Or was each research site treated as a single plot? How many soil samples were actually collected and analyzed, from each treatment and total for the study? What do you mean by "nine such replicates were maintained per soil"? Are you saying you broke the composite samples (from the 10 random cores) into 9 subsamples, or are you saying you collected 10 random cores from nine different sites at each of the study sites? Or maybe it is something else (I can come up with other possibilities). In short, the experimental design section needs to be expanded upon. I cannot picture the experimental design based on this description, and if readers cannot understand the experimental design nothing past this point matters, this manuscript is not publishable, because we cannot adequately evaluate the work and what the results actually mean.

Reply: As per the referee's comment, all the information will be corrected / provided in the revised MS. We have all the information pertaining to the soil sampling. We will provide much more information in the revised MS.

Line 96 – Were samples stored on ice while in the field during collection?

Reply: Yes. All the samples were stored in the coolant box; brought to the lab on the same day and stored at -20°C. We will include it in the revised MS.

Line 121 – The process used to establish the SBQ values is critical to evaluating this work, both for the reviewers right now and for future readers. Supplementary Table 1 is critical to this, and should therefore not be a supplementary table. The process of developing these threshold values needs to be clearly and completely described, therefore this table needs to be part of the manuscript.

Reply: As referee's suggestion, we will include the Supplementary Table 1 in the part of MS during revision.

Lines 145-146 – Where did the four soil samples from the three locations come from? I assume the three locations are the three study sites, but where do the four soil samples come from. See comment above about the need to clearly and completely describe the experimental design.

Reply: As replied earlier, we will include all these information more clearly and completely in the revised MS.

Lines 150-153 – Source(s) of these formulae?

Reply: Appropriate reference will be included.

Line 158 – Again, I think this table is too important to understanding the manuscript to be a supplemental table.

Reply: As per the referee's suggestion, we will include the Supplementary Table in the main manuscript during revision.

Lines 161-162 – Why is "high for variable a and low for variable b" scored 3 while "low for variable a and high for variable a" scored 2?

Reply: In quadrant plot, we always kept major contributor variable in 'x axis', while the secondary contributor is in 'y axis'. Hence, High x and low y is 3 and low x and high y is 2.

Lines 166-171 – Where did the farmers' fields come from? How and why were they chosen? How do they compare to the three study sites (cropping systems, soils, climate, etc., see Table 1)? You cannot just randomly drop another set of variables into the middle of a scientific paper and not explain them.

Reply: Farmers' fields are randomly chosen to check whether the SBQIs being worked in the fields and also to assess the relatedness among the SBQIs. We aware that this is not the final solution for SBQI. This exercise is purely to reconfirm the stability of SBQIs calculation and their inter-relation. After calculating the SBQIs from long-term manure experimental soils, in order to confirm their consistency, this is the only option available. From this part of the experiment, we proved that the SBQIs scaled from long-term manure experimental soils had same level of resolution and correlation in the farmers' soil.

Table 2 – Using a symbol such as an * would be a better way to indicate significance than bold values. Bold may not translate well if the paper gets copied in the future. Table 3 – The p values (where appropriate) are <0.001, not 0.000. Same comment in Line 182.

Reply: We will correct it as per the referee's remarks.

Section 3.3 – Details of the farmers fields have not been provided, so we do not know what we are making comparisons to here and in Table 6. I see the reference in Table 6 to supplementary materials, but this is critical information. It needs to be part of the paper, not part of the supplementary materials.

Reply: As replied earlier, SBQIs were measured in the farmers' soil in order to validate the SBQIs derived from the long-term manure soils. Hence, this is not the final conclusion about the farmer's soil quality. We will include it in the revised MS.

Line 251 – I have already discussed the need to better explain the experimental design, but four distinct soil samples? Really? You have three study sites with four treatments at each study site, shouldn't you have, at a bare minimum, 12 samples?

Reply: As per the referee's remarks, the soil sampling strategy will be thoroughly revised to understand better.

Lines 254-255 – Now different cropping sequences are being brought in. I don't remember any previous mention of different cropping sequences. Explanation of the experimental design for this project is extremely confusing!

Reply: We will revise to avoid all the confusions. As referee pointed out, we have included the crop sequence here which is unnecessary to the discussion.

Lines 258-260 – Don't need six citations here.

Reply: We will reduce it.

Lines 260-262 – Are there works from other research groups (other than the authors) that would indicate the results found here are reasonable? How about results from other environmental settings? SOIL is an international journal, to be publishable in SOIL this work needs to be tied into the bigger picture, the authors need to show why there should be international interest and not just interest within India.

Reply: Few works at different locations are available. We will include it during revision.

Lines 269-273 – This is just a repeat of what has already been said in the Materials and Methods. Not needed here.
Reply: We will correct it.

All the SQIs were found capable of separating the soils and treatments studied, and each has its own strengths and weaknesses. So, which of the SQI approaches do the authors recommend based on this work? Did on perform better than the others based on some objective method, was one better than the others because of its simplicity, or independence from "expert" (subjective) opinion, etc.? You get at this in the Conclusion, but seem to indicate SQI5 is the best after saying in Lines 330-331 that SQI5 needs more investigation/development.

Reply: As pointed out by referee, I would prefer SBQI5, which will provide more information. We will explain the merits and demerits of each method during revision.

Starting on Line 332 – As previously mentioned, more information on the farmers' fields, how they were chosen, what their soils and management are, etc. needs to be supplied earlier in the manuscript.

Reply: We will include it in the revised MS.

---

## Author Response (AR1)

**Reply to the topical Editor**

MS Title: Development of soil biological quality index for soils of semi-arid tropics Author(s): Selvaraj Aravindh et al. **MS No.: soil-2019-60**

We thank the topical editor for carefully considering our manuscript and the reply to the reviewers' comments. We have incorporated all the points as commented by two referees without any omission in the revised MS. The changes were made in MS Word file with 'track changing mode'. The major revisions include as follows:

- a. Included updated references (yellow highlighted in revised MS).
- b. English correction by native English speaker (Florida International University, Miami)
- c. All the required information about the experimental soils (Table 1 and Materials & methods)
- d. Sampling strategies followed during investigation
- e. Providing two tables from Supplementary materials to main text and change accordingly.

Apart from these, I hereby provide the reply (in red font) to the queries raised by the topic editor during the evaluation process.

TE query #1. Could you please provide a list of the updated references that you are going to add to the manuscript and how and where are you going to incorporate them?

**Reply: As per the anonymous reviewer #2, new and recent and updated references were replaced with old on in the Introduction. Those references were listed here:**

[revised manuscript text omitted]

TE query #2. Table 1. Can you specify which improvements are you going to make, or directly show the improved table?

Reply: As pointed out by the reviewer #2, the soil texture was corrected; soil order was removed; the varied nutrient managements in the long-term manure experiment were detailed in the materials and methods. The initial soil characters including pH, EC, soil organic carbon, available N, available P and available K were included. The table heading corrected as "Study area and soil characteristics". The final revised table is as follows:

| Table 1. Study area and soil characteristics |                        |                     |                     |  |  |  |
|----------------------------------------------|------------------------|---------------------|---------------------|--|--|--|
| Details                                      | Coimbatore             | Madurai             | Kovilpatti          |  |  |  |
| Centre                                       | TNAU, Coimbatore       | AC & RI, Madurai    | ARS, Kovilpatti     |  |  |  |
| Geographical coordinates                     | 11°N, 77°E             | 9.97°N, 78°E        | 09.12°N, 77.53°E    |  |  |  |
| Altitude                                     | 426 m                  | 147 m               | 106 m               |  |  |  |
| Max and Min temperature                      | 34.2°C and 20°C        | 32°C and 23°C       | 36°C and 29°C       |  |  |  |
| Annual rainfall                              | 670 mm                 | 1100 mm             | 730 mm              |  |  |  |
| Climate type                                 | semi-arid sub-tropical | arid sub-tropical   | semi-arid tropic    |  |  |  |
| Year of establishment                        | 1909                   | 1975                | 1982                |  |  |  |
| Test crop                                    | Maize – Sunflower      | Rice – Rice         | Cotton              |  |  |  |
| Cropping method                              | Irrigated              | Wetland             | Dryland             |  |  |  |
| Variables                                    | Nutrient management*   | Nutrient management | Nutrient management |  |  |  |
| Soil texture                                 | sandy loam             | sandy clay loam     | Clayey              |  |  |  |
| Soil classification                          | Typic Haplustalfs      | Typic Haplustalfs   | Typic Chromustert   |  |  |  |
| Initial soil characteristics                 |                        |                     |                     |  |  |  |
| рН                                           | 8.30                   | 7.1                 | 8.1                 |  |  |  |
| Electrical conductivity (dS/m)               | 0.25                   | 0.24                | 0.36                |  |  |  |
| Soil organic carbon (mg/g)                   | 2.90                   | 6.40                | 3.10                |  |  |  |
| Available N (mg/kg)                          | 145.0                  | 182.0               | 106.0               |  |  |  |
| Available P (mg/kg)                          | 4.8                    | 13.4                | 3.1                 |  |  |  |
| Available K (mg/kg)                          | 303.0                  | 275.0               | 546.0               |  |  |  |

\*The nutrient managements adopted in each site are described in Materials and Methods.

TE query #3. Can you please specify how are you going to improve the Study site and experimental design incorporating all the demands of the sampling strategy, explanations on farmer fields and the rest of demands related to study site and methods section?

Reply: This sub-head in Materials and methods had been revised thoroughly focusing the comments made by reviewer #2. The details of long-term nutrient managements adopted; varied levels of nutrients applied in three different sites; choice of four long-term nutrient management treatments in all the three sites and their brief description were included in the revised MS. All the doubts raised by the reviewer were incorporated in the revised MS. Similarly, the confusions in replicated samples (subsample / biological replicate) were also detailed with sufficient information.

TE query #4. Can you please rethink and explain how the non-normal distribution of data can affect statistical analysis?

Reply: I have discussed this issue with the Professor (Agricultural Statistics) of our institute. The normal distribution or skewed / bimodal distributions did not affect the statistical analysis, as the sampling was done in a long-term manure applied fields. A variable from this experimental plot which will be constant but, the same treatment at another location will vary (due to soil type) caused this skewing effect. Since the ANOVA and standard errors are discriminative with sufficient low p value, there won't be any issue in the statistical data. The description of his explanations are as follow:

Anova is not sensitive to moderate deviations from normality especially in simulation studies. The three sites of long-term manure experiments with constant application of four nutrient managements will discriminate each other may end with non-normal distribution. The non-normal distributions have shown that the false positive rate is not affected very much by this violation of the assumption (Harwell et al. 1992, Lix et al. 1996). This is because when you take a large number of random samples from a population, the means of those samples are approximately normally distributed even when the population is not normal.

[revised manuscript text omitted]

SOC – Soil organic carbon; MBC – Microbial biomass carbon; SLC – Soil labile carbon; SPI – Soil protein index; DHA – Dehydrogenase; SIR – Substrate induced respiration. Threshold values are scaled as soil index scale ranged from 1 to 4 based on the literatures.

Table 3. Pair of variables used for the quadrant plot and their mean and regression coefficient (R2)

| Variable-1          | Variable-2          | Mean of variable -1 | Mean of variable - 2 | R 2 | P         |
|---------------------|---------------------|---------------------|----------------------|-----------------------|------------------|
| (major       | (secondary   |                     |                      |                       |                  |
| contributor) | contributor) |                     |                      |                       |                  |
| SOC          | MBC          | 7.29         | 382.51        | 0.237          | <0.001 |
| SOC          | SLC          | 7.29         | 480.30        | 0.417          | <0.001 |
| SOC          | SIR          | 7.29         | 3.20          | 0.409          | <0.001 |
| MBC          | SPI          | 382.51       | 5.46          | 0.089          | <0.001 |
| MBC                 | DHA                 | 382.51       | 11.51         | 0.259                 | <0.001 |
| MBC          | SIR          | 382.51       | 3.20          | 0.337          | <0.001 |

SOC – Soil organic carbon; MBC – Microbial biomass carbon; SLC – Soil labile carbon; SPI – Soil protein index; DHA – Dehydrogenase; SIR – Substrate induced respiration.

Table 23. Correlation coefficient (Pearson, n-1) of the observed variables from long-term nutrient management soils

| Variables | SOC           | MBC           | SLC           | SPI           | DHA           | SIR           |
|-----------|---------------|---------------|---------------|---------------|---------------|---------------|
| SOC       | 1.00 * |               |               |               |               |               |
| MBC       | 0.93 * | 1.00 * |               |               |               |               |
| SLC       | 0.74 * | 0.85 * | 1.00 * |               |               |               |
| SPI       | 0.68 * | 0.51          | 0.10          | 1.00 * |               |               |
| DHA       | 0.65 * | 0.81 * | 0.95 * | 0.05          | 1.00 * |               |
| SIR       | 0.80 * | 0.89 * | 0.93 * | 0.25          | 0.85 * | 1.00 * |

SOC – Soil organic carbon; MBC – Microbial biomass carbon; SLC – Soil labile carbon; SPI – Soil protein index; DHA – Dehydrogenase; SIR – Substrate induced respiration. <del>Values in bold are different from 0 with a significance level\* Correlation is significant at -p=the</del> 0.05 level-.

Table 34. Regression analysis of soil variables assessed for long-term nutrient management adopted soils

[revised manuscript text omitted]

 645
 of
 soil
 biological
 quality
 index
 calculated
 for
 the
 soil
 samples.

 \* Correlation is significant at the 0.05 level.

 samples.

---

## Author Response (AR2)

**Reply to the topical Editor**

MS Title: Development of soil biological quality index for soils of semi-arid tropics
Author(s): Selvaraj Aravindh et al.
**MS No.: soil-2019-60**

Comments to the Author:

Dear authors, your effort in the review process is highly appreciated. Thank you very much for this. In order to continue with the review process we need a further improvement in the English text and grammar correction by an editing service or by a native English colleague, as one of the referees has again requested. We really hope that you can provide that and we can move on with the review process. I encourage you to go on and thank you very much in advance for all the efforts.

Reply: As pointed out by one of the reviewer and the technical editor, we have improved the English of the text as well as the grammar correction. The improvement was kindly done by native English speaking colleague (Florida International University, Miami).

[revised manuscript text omitted]

---

## Author Response (AR3)

**Reply to the topical Editor**

MS Title: Development of soil biological quality index for soils of semi-arid tropics
Author(s): Selvaraj Aravindh et al.
**MS No.: soil-2019-60**

Comments to the Author:

We acknowledge the effort made to revise the English text and to address all referee's comments. The manuscript was reviewed a second time by one of the referees and there are some minor issues that need to be addressed. He also strongly recommends again to have a look to the English text. Could you please find a native independent English speaker to review critically the English text again? I encourage you to go on and address those minor comments, as well as a revision of the language, to complete a successful publication process.

Reply: As pointed out by the reviewer and the technical editor, I have made all the minor corrections as well as the English of the text. We have revised the English based on the native English-speaking colleague from Florida International University, Miami.

**Reviewer's suggestions as minor revision:**

In section 2.4, I find the formulas for calculating SBQI a bit confusing. The reason is the first SBQI is shown as "SBQI – 1 =", the second as "SBQI – 2 =", etc. I think the authors mean the first SBQI, second SBQI, etc. as they write this, but at first I was wondering why the first equation subtracted 1 from the SBQI, the second subtracted 2 from SBQI, etc. I recommend writing these as SBQI1, SBQI1, etc., or something similar, so it does not look like a mathematical operation is taking place on the left side of the equation.

Reply: As pointed out by reviewer, the SBQI-1 leads to confusion as 'subtraction'. Hence, the expression of different SBQIs was rewritten as SQBI1, SBQI2, SBQI3, SBQI4 and SBQI5. This change was made throughout the text, table and supplementary materials.

Line 218 – "significantly discriminated against the soils". I think the authors are saying here that the SBQI-1 method was able to differentiate between soils. Discriminated against carries a negative connotation and I do not think it represents what the authors were trying to say. Same comment in lines 231-232 and other places where this wording occurs.

Reply: As per the suggestion made by the reviewer, the phrase 'discriminated against' was changed throughout the text.

Line 339 and Line 341 – Both these lines say the 25 farm fields were used for validation, this is repetitive.

Reply: The repetitive phrase was removed in the revised MS.

Lines 350-356 – None of these issues were tested during the study (or if they were, it was not described in the manuscript). Therefore, this comes across as speculation.

Reply: The issued listed in this part of discussion is purely based on the previous works and reports, not from this study. These assumptions may lead to speculation. Hence, we have removed the lines in the revised MS and concluded accordingly.

Lines 364-365 – "This method also identifies the constraints of the soil biology better than the other four methods." I don't see how this was accomplished in this study. What I see in this study is the idea for a new SBQI, and the tests done showed that this SBQI gives comparable results to other, already established SBQI. Therefore, it appears to be a valid way to determine SBQI. Further research is needed to determine any benefits and drawbacks of this method as compared to other methods.

Reply: This is in response with the earlier comment too. We listed the possible constraints due to low SBQI at the end of the discussion (Lines 350-356 of revision-2). However, we have deleted it in this revision as per the advice of the reviewer. Hence, this sentence also be removed accordingly.

[revised manuscript text omitted]

---

## Author Response (AR4)

**Reply to the Topical Editor**

MS Title: Development of soil biological quality index for soils of semi-arid tropics
Author(s): Selvaraj Aravindh et al.
**MS No.: soil-2019-60**

Comments to the Author:

Thank you for the revised manuscript. I am happy to inform you that your paper is now accepted subject to technical corrections. The topical editor has identified several language issues that need to be resolved and has detailed this. Please carefully consider this. I also think the manuscript will benefit from a thorough copy editing. Copernicus provides this service and I suggest that you interact with Copernicus to further improve the language.

Reply: As pointed out by the reviewer and the technical editor, I carefully scrutinize the English corrections pointed out by the topical editor. I express my sincere thanks to Topical editor who improved the manuscript to a good shape.

**Topical editor's suggestions:**

1. I think the title (Development of soil biological quality index for soils of semi-arid tropics) needs an "a" before "soil biological quality index", it would be better as "Development of a soil biological quality index for soils of semi-arid tropics". Could you please check with your native English translator?
Reply: Modifying the title as "Development of a soil biological quality index for soils of semi-arid tropics" looks better. Hence 'a' was added in the title.

2. Can you substitute the "the" of line 10 "the soil quality index" by an "a soil quality index"?
Reply: Corrected as 'a'

3. In the discussion could you please change the present perfect forms by a simple past forms? Such as:
Line 255 "we have developed" by "we developed"
Reply: Corrected
Line 256 "have chosen" by "chose"
Reply: Corrected
Line 308 "we have adopted" by "we adopted"
Reply: Corrected
Line 311 "we have not picked" "we did not pick"
Reply: Corrected
357 "we have investigated" by "we investigated"
Reply: Corrected
Line 316 "can able" is not correct, can you replace it by "was able"?
Reply: Corrected

Apart from these, the manuscript is thoroughly scrutinized and changed as simple past form of verbs.

4. I am not sure if those sentences are correctly formulated:
Line 317 "relating to soil functioning" do you mean "and relate it to the soil functioning" ? If that is so, can you change the sentence?
Reply: Corrected as 'and relate it to the soil functioning'

Line 338 "Twenty five farmers' field in and around…, have assessed, …" I think you mean "….. were assessed…"
Reply: Corrected as 'were'

Line 355 "these soils may deter their quality …." change by "the quality of soils may decrease and may be reflected in declining crop productivity"
Reply: As per the suggestion, the sentence was rephrased.

5. Line 363-364
Can you correct this sentence "as it is with less statistics and a more biological approach" by "as it implies less statistics calculations and it is more based on a biological approach"
Reply: As per the suggestion, the sentence was rephrased.